# Skin Microbiota: Setting up a Protocol to Evaluate a Correlation between the Microbial Flora and Skin Parameters

**DOI:** 10.3390/biomedicines11030966

**Published:** 2023-03-21

**Authors:** Paola Perugini, Camilla Grignani, Giorgia Condrò, Harald van der Hoeven, Annamaria Ratti, Antonella Mondelli, Antonio Colpani, Mariella Bleve

**Affiliations:** 1Department of Drug Sciences, University of Pavia, Via Taramelli 12, 27100 Pavia, Italy; 2Etichub, Academic Spin-Off, University of Pavia, Via Taramelli 12, 27100 Pavia, Italy; 3CLR—Chemisches Laboratorium Dr. Kurt Richter GmbH, Sperenberger Straße 3, 12277 Berlin, Germany; 4Bregaglio, Via Trento e Trieste, 97, 20853 Biassono, Italy; 5Kelisema, Via Urago 13b, 22038 Tavernerio, Italy; 6I Beauty, Via G. Donizetti, 109, 24030 Brembate di sopra, Italy; 7Department of Management, Information and Production Engineering, University of Bergamo, Via Salvecchio 19, 24129 Bergamo, Italy

**Keywords:** skin microbiota, biophysical parameters, *Cutibacterium acnes*, skin dysbiosis, in vivo protocol

## Abstract

The concept of skin microbiota is not really clear and more accurate approaches are necessary to explain how microbial flora can influence skin biophysical parameters in healthy individuals and in pathology patients with non-infectious skin disease. The aim of this work is to provide a suitable, fast and reproducible protocol to correlate skin parameters with the composition of skin microbiota. For this purpose, the work was split into two main phases. The first phase was focused on the selection of volunteers by the administration of a specific questionnaire. The skin microbiota was then collected from the forehead of selected volunteers as a test area and from the shoulder as control area. On the same skin area, the biophysical parameters, such as trans-epidermal water loss (TEWL), sebum level (SL), porphyrin intensity, keratin content and stratum corneum water content were taken. All parameters were taken at t0 and after 15 days without changes in the volunteers’ lifestyle. A strong correlation was found between forehead and shoulder area for porphyrin intensity, pH and TEWL parameters, and between *Cutibacterium acnes* and some biophysical parameters both in the forehead and the shoulder area. The procedural setup in this work represents the starting point for evaluating problematic skins and the efficacy of cosmetic products or treatment against skin dysbiosis.

## 1. Introduction

Lately, the concept of microbiota is emerging both in pharmaceutical and cosmetic fields, and the terms are often used incorrectly. For example, the terms microbiota and microbiome are often confused. The entire set of microbes present in a specific habitat is called the microbiota. This term includes not only bacteria but also viruses, fungi, yeasts, and mites associated with a specific body area such as nose, gut, oral mucosae and skin. The microbiome, on the other hand, outlines the genetic material (DNA/RNA) and genome of these microorganisms [1,2].

While the intestinal microbiota is largely known and rather well understood, the concept of skin microbiota is a quite recent one and much knowledge is still needed to obtain a clear and complete picture of the microbial situation that populates our skin [3,4]. The choice to focus on the skin microbiota originates in the skin’s ability to represent an interface between the environment and the body, as well as in the fact that the skin is one of the largest human organs [5,6]. The skin represents the first line of defense and barrier against different factors, such as pathogens, UV light, chemicals, and mechanical injury. Thanks to corneocytes and lipids present on the surface layer, it normalizes temperature and the amount of water.

Like other tissues that interact with external habitat, the human skin is an ideal location for the controlled growth of bacteria. In fact, a single square centimeter contains up to one million microorganisms [7,8]. On the surface of the skin there are commensal and pathogenic bacteria. Commensal bacteria, which can be transient or resident, can create a healthy, balanced ecosystem, whereas pathogenic bacteria can invade the tissue, causing harm and irritation. Between these microbial communities and the host tissue, various relationships can be established, some of which bring benefits while others can have negative effects, by altering its composition and determining the onset of certain non-infectious skin diseases such as acne vulgaris, atopic dermatitis and rosacea [3,9,10].

Maintaining the balanced composition of microbiota thus seems essential for preserving skin health and beauty. The main aim of several studies on skin microbiota was to understand how microbial composition affects skin condition and under what conditions the invasion of harmful communities can occur [11,12,13,14]. In fact, research has shown that the imbalanced composition of microbiota led to epithelial dysfunction and changes in immune regulation, with the consequence of the colonization of pathogenic microbial populations [15,16].

With the advent of new sequencing techniques, scientists were able to classify the skin bacteria into the following phyla: *Actinobacteria, Bacteroidetes, Firmicutes,* and *Proteobacteria* [17]. Moreover, they were able to divide the microbial population based on the skin area types: sebaceous or oily (face, chest and back); moist (bend of elbow, back of knee and groin) and dry (volar forearm and palm) [18,19,20]. For example, in moist areas, *Staphylococcus* and *Corynebacteria* species prevail; in sebaceous ones, lipophilic species, such as *Cutibacterium,* which lives in anaerobic and lipid-rich environments, are present. In dry areas we find a large amount of *Staphylococcus, Micrococcus, Streptococcus* and *Corynebacterium.* Furthermore, at a microscopic level, there are even smaller habitats corresponding to the eccrine and apocrine glands, to the sebaceous glands, and to the hair follicles, which have their own characteristic microbiota [3,4,21].

The composition of skin microbiota is not fixed and like DNA, the composition of microbiota has an individual signature. It varies significantly from individual to individual and is influenced by intrinsic and extrinsic factors [3,22].

To understand the role of the skin microbiota in the onset or treatment of certain pathological conditions and its influence on the wellbeing of the skin, one approach could be to evaluate the relationship between skin biophysical parameters and the microbial population. A recent systematic review of this topic reported results from research carried out in the main scientific databases [23]. The analysis of 15 papers on acneic skin and 12 articles on psoriasis conditions did not find a correlation between skin microbiota alteration and biophysical parameters, probably due to the extreme variability among the subjects involved. On the other hand, some changes in certain skin physiology parameters seem to occur in patients with atopic dermatitis, such as the decrease in stratum corneum water content and a certain increase in skin pH and in TEWL parameters [23].

However, before starting to investigate this correlation in skin disorders, it is necessary to evaluate these parameters in healthy individuals and to search for a correlation between these parameters and the microbial population. Skin hydration is essential for ensuring an adequate colonization of microorganisms. The literature shows that skin with a high degree of hydration allows the development of a richer and more abundant bacterial flora compared to dry skin [24]. A lower degree of hydration corresponds to a lower bacterial diversity and a greater risk of colonization by pathogenic bacteria. A low degree of skin hydration also leads to a greater transepidermal water loss with alteration of the skin barrier and roughness. For this reason, the measurement of this parameter could be used to understand the level of skin hydration that gives people a balanced microbial composition and thus healthy skin [21].

As mentioned above, in some habitats the presence of sebum is fundamental for determining the composition of the bacterial flora. In fact, if sebum levels increase, the presence of *Cutibacterium* species, including *Cutibacterium acnes*, increases. It is important to detect the presence of this strain, because it is a commensal bacterium that, in some cases and for some phylotypes (IA1 for example identified in a phylogenetic tree) determines the emergence of acne vulgaris [24]. *Cutibacterium acnes* strains were precisely classified into several subtypes subdivided into six main phylotypes: IA1, IA2, IB, IC, II and III [25]. This bacterium is able to produce some catalytic factors called porphyrins, which in Wood’s light (a wavelength between 370 nm to 450 nm) emits red-orange fluorescence. In this way, the presence of this strain on the skin can be detected. The intensity of the fluorescence of these chromophores is proportional to the density of *C. acnes* [26,27].

However, in order to correctly analyze skin microbiota, it is necessary to choose the most suitable sampling method. In the literature, the main methods used involve microbial culture, cotton swabs, tape stripping, or skin biopsies. [28]. However, skin biopsy is an invasive method. Between swabs and tape stripping, swabs are a less reliable and accurate method, because some parameters, such as pressure, number of sampling times, direction and the material of the swabs, are not standardized, and they could not be controlled [29]. On the other hand, tape stripping (TS) is a well-known, established method, but it could lead to misinterpretations due to the interference signals caused by components in the stratum corneum or by the skin-rinsing method. [30].

The general aim of this paper is to provide a suitable, fast and reproducible protocol to correlate skin parameters with the composition of skin microbiota. For this purpose the work was split into three main phases. The first phase focused on the selection of healthy volunteers by administering a specific questionnaire, on intrinsic and extrinsic factors, to obtain a homogeneous panel.

The second phase focused on the sampling of skin microbiota using a simple, rapid method without undesirable interferences.

Finally, in the third phase of this work, biophysical parameters were collected, such as transepidermal water loss (TEWL), sebum level (SL), porphyrin intensity, keratin content, stratum corneum water content and skin microbiota in two different body areas, the forehead and the shoulder.

In order to ensure that the analysis is reproducible and the daily routine of volunteers did not change skin microbiota, the parameters were taken at t0 and after 15 days without changes in the volunteers’ lifestyle.

## 2. Materials and Methods

### 2.1. First Phase: Volunteer Selection

The first part of this phase was focused on the selection of volunteers to obtain a homogeneous panel. For this reason, a preliminary step was conducted through the compilation of a questionnaire with specific criteria.

A total of 229 volunteers, ranging between 19 and 50 years old, were recruited. All volunteers signed a consent form allowing us to treat their personal data, according to Italian law (GDPR 2016/679).

Figure 1 shows a flowchart describing the primary and secondary exclusion criteria chosen to select the candidates. The selection of these parameters is correlated with the possibility of obtaining variable data in terms of flora composition.

### 2.2. Design of Experiment

A total of 20 healthy female volunteers were selected from the first phase, as described above. For each volunteer, a test area (forehead) and a control area (shoulder) were defined.

Assessments of both skin biophysical parameters and skin microbiota were performed at the beginning of the study (T_0_) and after 15 days (T_1_) without any changes in the volunteers’ lifestyle.

### 2.3. Acquisition of Skin Biophysical Parameters

The in vivo study was carried out according to the Helsinki declaration (Ethical Principles for Medical Research Involving Human Subjects) [31].

All measurements were made in an air-conditioned room with controlled temperature and humidity (T = 22 °C, relative humidity [RH] = 70 ± 5%). Subjects were preconditioned in the room for at least 15 min before the measurements were made.

The instruments used in the evaluation of skin parameters involved contact between the skin and a series of probes that did not cause discomfort, pain, or damage the skin.

The skin parameters investigated in the present study were the stratum corneum water content (electrical properties of skin), pH measurement of the skin surface, morphological skin analysis by determining pore size and amount, intensity of porphyrins, total quantity of sebum, transepidermal water loss (TEWL), and the amount of keratin protein in the stratum corneum (SC). Skin parameters were evaluated using an MPA 580 multiprobe adapter system cutometer (Courage&Khazaka electronic GmbH, Cologne, Germany) equipped with various probes. The stratum corneum water content (SCWC) was assessed with a CM 825 corneometer probe. Corneometry is a technique used to assess the hydration of the outer layer of the epidermis, which is known as the stratum corneum [32]. Since skin is a dielectric medium, all variations in hydration result in a corresponding change in the skin’s electrical capacity [33]. The device used in the present trial was equipped with a 49 mm^2^ surface probe that allows precise measurement in 1 s within a depth range of 10–20 μm into the stratum corneum. The parameters were expressed using an arbitrary score scale (0–100 AU).

Skin barrier integrity was assessed by measuring transepidermal water loss (TEWL) using a TM 300 Tewameter probe [34]. TEWL was quantified in g/m^2^h by a skin evaporimeter made of a small cylindrical open chamber (1 cm in diameter, 2 cm in height) with two hygrometric sensors connected to a microprocessor plugged into a computer workstation. The device allows the recording of TEWL values (ranging from 0 to 90 g/m^2^h) as well as the relative humidity (ranging from 0% to 100%) and probe temperature.

The pH of the skin surface was measured with a pH-meter PH 900 equipped with a planar probe head which combines the H+ ion sensitive electrode and the reference electrode in one rod. The quantity of sebum is measured by a Sebumeter SM 815 probe, constituted by a tape that is put into contact with skin. The tape becomes transparent in relation to the amount of sebum and the transparency is measured by a photocell. The light transmission reflects the sebum content.

The stratum corneum protein content (PC) was investigated through tape stripping using a D-squame device [35]. This method involves the application of adhesive tape with a diameter of 14 mm to the skin and its subsequent removal to strip off a layer of the stratum corneum. A constant pressure of 225 g/cm^2^ is impressed on the disk surface. The infrared densitometry (IRD) technique (850 nm wavelength) was used to quantify the absolute amount of stratum corneum removed by each tape strip. The results are expressed as PC in µg/cm^2^.

Visiopor PP34 with a specific UV light was used to visualize the fluorescence of the pores and porphyrins in a skin area of minimum 8 × 6.4 mm^2^. The intensity of follicular fluorescence and the extent of area involvement are proportional to the porphyrin content [36].

### 2.4. Skin Flora Collection

The tape stripping method using adhesive D-100-D-Squame Stripping Discs on a polyester carrier sheet (22.0 mm ∅) (D-squame^®^, Clinical and Derm, 12221 Merit Dr Ste 940 Dallas, TX 75251 United States) was used for collecting skin microflora [37].

Before proceeding with the microbiological sampling, the laboratory was sanitized and, moreover, people not involved in the work were forbidden to enter. All the instruments used were therefore sanitized and the microbiological analysis was conducted under a laminar flow hood. The sanitization of surfaces and instruments was always performed at the end of the analysis of each subject.

Preliminary analyses were made to identify the sources of contamination and take steps to restrict or eliminate the contamination:Contamination of the strip: one strip was removed from the film to which it is attached using tweezers. The strip is then inserted directly into a sterile Eppendorf (B12020). This serves to check for possible interference with bacteria present on the tape at purchase.Environmental contamination: a second strip, removed from its film as described above, was left in a laboratory environment for 24 h to observe possible interference from bacteria present in the air (B22020).Device contamination: the third preliminary analysis involved the application of a constant pressure of 10 s with the appropriate cylinder on one other strip lying on the film, which was then placed in a sterile Eppendorf (B32020). This was useful to verify the possible interference with bacteria present on the cylinder used. One more analysis was performed on another strip, placed on its film, pressured for 10 s with the cylinder, and then placed inside the IR instrument to mimic the reading of the amount of keratin. It was subsequently inserted into a sterile Eppendorf (B42020). This procedure allowed us to understand the possible interferences with bacteria using the device to measure the microbial and the keratin content on the same strip.

#### 2.4.1. In Vivo Skin Microbiota Preliminary Tests

Several preliminary tests were carried out in order to define the best in vivo procedure to collect the real skin microbiota, avoiding as much as possible any external contamination.

First of all, three adjacent zones (A, B, C) both on the forehead (F) and on the shoulder (S) of three volunteers were identified:Zone A: Samples were collected separately using 5 strips for each area (from C1FA to -C5FA for the forehead and from C1SA to -C5SA for the shoulder). The objective was to understand if it is possible to obtain a quantity of genetic microbial material to allow the analysis of the microbiota in all 5 first strips of skin.Zone B: Samples were collected using 5 strips and examined together in a single analysis. The objective was to understand if it is more effective to analyze the 5 strips together in order to reduce the number of total analyses.Zone C: The skin areas were cleaned 2 h before sampling.

#### 2.4.2. In Vivo Skin Microbiota Sample Collection

The sampling procedure was performed in a similar way on two skin areas: forehead (test area) and shoulder (control).

The in vivo skin microbiota analysis was performed on 20 healthy female volunteers selected from the first phase. The procedure was set as follows (Figure 2):Removal of the adhesive disks (tape) from the film on which they were placed using tweezers.Application of the adhesive disk on the skin under constant pressure using the cylinder for 10 s.Removal of the adhesive pads from the skin using tweezers.Determining the SC protein content with IR.Placing the strip into a sterile Eppendorf and then closing the tube.This procedure was repeated three times on the same area of analysis.

### 2.5. Microbiota Analysis by Sequencing:

The bacterial composition analysis was performed with the Illumina platform with a MiSeq^TM^ instrument (IGA Technology Services Srl, Udine, UD, Italy). DNA was isolated from patches using a NucleoSpin Tissue Mini Kit for DNA from cells and tissue (Macherey-Nagel GmbH & Co., Valencienner Str. 11, 52355 Dueren, Germany, Germany). Libraries were prepared by following the Illumina 16S Metagenomic Sequencing Library Preparation protocol in two amplification steps: an initial PCR amplification using locus-specific PCR primers and a subsequent amplification that integrated relevant flow-cell binding domains and unique indices (NexteraXT Index Kit, FC-131-1001/FC-131-1002, Bio-Active Company Limited, 188/1 Chuea Phloeng Rd, Chong Nonsi, Yan Nawa, Bangkok 10120, Thailand). Primer sequences used to amplify both the variable 16S (V3-V4) and ITS regions followed: 16S-341F - CCTACGGGNBGCASCAG -3_’ and 16S-805R 5_’- GACTACNVGGGTATCTAATCC -3_’. Libraries were sequenced on a MiSeq instrument (Illumina) in paired end 300-bp mode read length.

### 2.6. Statistical Analysis

A statistical analysis of the data collected from the in vivo study was performed. The comparison of the data collected over time was performed using a *t*-test. A significance of 95% was chosen; thus, changes were considered significantly different when the *p* value was <0.05.

The Spearman’s rank correlation test was used to evaluate the correlation between two variables [37]. It has a value between +1 and −1, where 1 is a total positive correlation and −1 is a total negative correlation. The correlation reflects the noisiness and direction of a relationship, but not the slope of that relationship, nor many aspects of nonlinear relationships. There is no general consensus on the classification of the relationship for different coefficients. In this work, the criteria reported in Table 1 were applied.

## 3. Results

The main aim of the present work was the development of an in vivo protocol for instrumental and microbiota analysis on the skin, using the forehead as test area and the shoulder as control.

### 3.1. In Vivo Study First Phase: Volunteer Selection

In the volunteer-selection phase, after identifying the main exclusion criteria that affect the skin microbiota (Figure 3a) additional criteria, considered secondary, were applied to obtain an even more homogeneous panel of volunteers (Figure 3b).

Following an initial analysis that was focused on the primary inclusion criteria, 162 subjects were eliminated. These subjects were not suitable for participation in the study because they met the exclusion criteria, i.e., factors which the literature identifies as being responsible for significant changes in the bacterial flora of the skin. Women using hormone therapy or contraceptive methods, subjects who have chronic diseases and regularly take medication, smokers and people who live with a pet (dog and/or cat) were excluded from the study.

To obtain a strictly female panel, all men were excluded. Furthermore, since hormonal conditions can also be a reason for variation in the bacterial flora, menopausal women were also excluded, and only fertile-age women were selected. These are factors that, although to a lesser extent than previously seen, can affect the composition of the skin microbiota. The application of these last criteria led to the elimination of 40 more subjects. The 27 remaining subjects were then categorized according to age and cosmetic routine in order to better understand the panel. A total of 63% of the women were between 20 and 30 years old, 18% 30–40 years old and 19% between 40 and 55 years old. Subsequently, the cosmetic routine of these 27 women was assessed. Frequency and routine of cleansing the face and the use of make-up were evaluated. From these further analyzes, a panel of 20 subjects with very similar cosmetic routine was assembled.

### 3.2. Instrumental Analysis

The in vivo skin biophysical parameters considered in this study are: stratum corneum water content, TEWL, pH, sebum levels, and the amount of keratin and porphyrins. Table 2 shows the results obtained for the forehead and shoulder at t0 and after 15 days. The average of the values of the 20 selected subjects with the standard deviation was reported.

The results proved that skin parameters remain quite constant over time for the panel of volunteers

Moreover, focusing on the two areas investigated, the values of stratum corneum water content, TEWL, pH, porphyrin intensity and count and protein content in the SC are quite homogeneous. The only results differing in the two areas are the level of sebum and the porphyrin count and size. In fact, the shoulder shows a lower quantity of sebum 15.85 ± 9.16% (shoulder t0), unlike the forehead (forehead t0) 81.7 ± 36.35%.

Table 3 shows the statistical analysis as carried out on biophysical parameters. Spearman’s correlation rank test highlights strong negative correlation between porphyrin intensity and the protein content, sebum and TEWL. Furthermore, a strong correlation between forehead and shoulder area was recovered for porphyrin intensity, pH and TEWL parameters.

### 3.3. Microbiota Analysis

#### 3.3.1. Preliminary Analyses

Table 4 shows the data obtained from the first preliminary analyses made to identify the contamination of strips. In particular, the most present bacterial phylum level is recovered on the strip (Sample B12020). To check the environmental contamination and contamination during the sampling procedure, phylum levels above 1% were considered.

Actinobacteria, found at a higher percentage in sample B12020, are also present in all strips. In fact, Actinobacteria turn out to be one of the most extensive phyla in the bacterial domain, and are found in several environments, from terrestrial to aquatic [38,39,40]. As a result, the presence of this high percentage can be correlated with environmental contamination.

However, in the last sample, which represents the whole procedure, the percentage of the Actinobacteria phylum turns out to be much lower, and consequently the right sampling procedure was confirmed.

The Cyanobacteria phylum is present in greater quantities in sample B22020, as it is associated with environmental bacterial contamination. In fact, Cyanobacteria are photosynthetic microorganisms that inhabit a wide range of both aquatic and terrestrial environments. However, they are not part of the skin microbiota [41].

#### 3.3.2. In Vivo Skin Microbiota Preliminary Tests

Figure 4 shows the mean results of bacterial species level obtained from five strips separately collected and analyzed, taken from the forehead and the shoulder A areas, from three volunteers.

Results obtained from these preliminary analyses highlighted that the first strip presented greater environmental contamination from Staphylococcus and was therefore discarded. Results collected in zone B and zone C were almost the same (data not reported).

#### 3.3.3. In Vivo Skin Microbiota Evaluation

The in vivo study was performed on the same volunteer panel previously selected (see Method Section 2.1).

The final procedure to perform in vivo study involved the collection of two strips from each subject starting from the second one. For each volunteer, two strips were taken on the forehead and shoulder area, obtaining two samples. They were then analyzed together, obtaining a cumulative analysis. Table 5 shows mean bacterial species distribution obtained analyzing two strips together (strip 2 and strip 3, C2- 3) both from the forehead area (FA) and the shoulder area (SA).

Results confirmed that the method applied by analyzing only the second and the third strip together yielded enough genetic material for the investigation of the skin bacterial species-level profile distribution.

Table 6 reports the results of the microbial distribution in terms of principal phyla on the forehead and shoulder over time obtained from 20 volunteers. The phyla taken into consideration in this work were: *Actinobacteria, Proteobacteria, Bacteroidetes, Cyanobacteria* and *Firmicutes.*

The analyses of the bacterial profile distribution show homogeneity for both areas of the body.

Only the Cyanobacteria population shows a nonhomogeneous distribution over time. However, this could be air contamination, because this phylum is not well correlated with skin microbiota

Figure 5 shows the results obtained from the sequencing analysis of the microbiota at the species level during in vivo study carried out on 20 volunteers. In particular, the most representative species are compared at t0 and after 15 days without any change in the volunteers’ lifestyle.

Figure 6 shows statistical results of the analysis of distribution of the main bacterial species recovered on the skin. Results showed a strong positive correlation between *Cutibacterium acnes* presence in the forehead and in the shoulder (r = 0.55). In the shoulder area, a negative strong correlation was found between *Cutibacterium acnes* and *Staphylococcus* species (r = −0.42) and a positive moderate correlation between *Corynebacterium* species and *Staphylococcus* species (r = 0.35).

Table 7 shows Spearman’s rank correlation coefficients obtained by analysis, as well as microbial species and biophysical parameters both in the forehead and in the shoulder areas.

Results highlighted a strong positive correlation between *Cutibacterium acnes* and biophysical parameters both in the forehead and the shoulder area.

## 4. Discussion

The concept of skin microbiota was practically unknown until the early 2000s, and up to now only very few scientific papers have reported on correlations between the composition of the cutaneous microbiota and skin biophysical parameters. This work aims to provide a scientific protocol in order to investigate the microbial bacterial distribution on skin in correlation with biophysical skin parameters.

The microbiota is very different among individuals, depending on intrinsic and extrinsic factors. In the present work, extrinsic factors, including normal living habits such as occupation, the environment in which the individual lives, the presence of pets and diet, were investigated through a questionnaire. Another factor in determining bacterial variability could be the use of disinfectant in clothes washing.

The main focus of this work was to maximally limit individual variability, which has been shown to be one of the most important challenges in in vivo skin microbiota evaluation.

The new approach to the recruitment of volunteers on the basis of a specific questionnaire permitted us to obtain a very similar panel of subjects in terms of lifestyle.

The skin microbiota sampling method chosen demonstrated that with only three uses of tape stripping, the bacteria on the skin can be collected and analyzed, using a very fast and non-invasive method, thus avoiding any invasive method such as skin biopsy and increasing the compliance of volunteers.

The skin biophysical parameters were also investigated in order to demonstrate that they remain constant in a homogenous panel of volunteers over 15 days without changes in lifestyle.

The simultaneous analyses carried out on the forehead and on the shoulder aimed not only to evaluate the composition of commensal bacteria present on healthy skin but also to verify the possibility of keeping an area as a monitor for individual variability. The skin area tested was the forehead, because a great many studies concerning microbiota were focused on face microbial dysbiosis. Furthermore, a control area for each volunteer was defined and investigated. In fact, in this way it would be possible to control over time the parameters defined as a baseline for each volunteer and then to investigate the modification of biophysical parameters and microbiota in the area with skin alteration. In this way, it is possible to observe how they vary according to a particular skin disease and according to its severity index.

The shoulder was chosen as a control area for the forehead area because a similar skin microbial composition has been reported in the literature [7]. A strong correlation was found between the forehead and shoulder areas for porphyrin intensity, pH and TEWL parameters, showing that these parameters in the shoulder area could be used as an intraindividual control during a clinico-instrumental evaluation study for a face treatment.

The main phyla present on the skin, as reported in the literature, are Actinobacteria, Proteobacteria and Firmicutes. From the data obtained through the analyses of microbiota, it is observed that the species most represented are *Cutibacterium acnes, Staphylocuccus aureus* and *Corynebacterium species*. *Cutibacterium acnes* on the skin of healthy volunteers represents the commensal species that help to establish the correct balance of microbial flora. The other species can be considered secondary in characterizing the skin microbiota. Even for this data, the shoulder had demonstrated correlated analyses.

A new and very important result obtained from this work was the strong positive correlation between *Cutibacterium acnes* and some biophysical parameters both in the forehead and the shoulder area. This statistical correlation has never before been reported in in vivo study in literature. However, it is important to take into account these results, especially when evaluating a specific skin disease, such as acne vulgaris. In fact, it is likely that, in acneic skin, porphyrin quantity, intensity and size will be much higher, as more bacteria will be present, producing more porphyrins, more irritation and inflammation, and also leading to a growth of pores, which will be filled with pus and an excess of sebum.

Finally, the study also showed that biophysical parameters as well as microbial composition did not change over 15 days, without any change in the lifestyle of the volunteers. These results are very important, because it is well known that healthy skin is able to maintain microbial distribution, as well as that of some biophysical parameters, on an even level over time. Thus these results demonstrated that the study was properly carried out and that the same procedure would be employed to evaluate any case of dysbiosis.

## 5. Conclusions

This work demonstrated that an in vivo protocol for testing both skin microbiota and biophysical parameters was successfully obtained.

The procedure described in this work, based on a specific questionnaire and analyses of biophysical parameters as well as of skin microbial distribution, yielded reliable and reproducible data.

With the new protocol proposed in this work, it is possible to collect bacterial flora on healthy individuals without an invasive method. Moreover, it was demonstrated that, with the forehead as test area, the shoulder is an adequate control area in the investigation of microbial composition and biophysical parameters.

Thus, this method accurately shows the capability of healthy skin to maintain skin-barrier function, hydration and a commensal bacteria colonization over time, avoiding dysbiosis, when any change in lifestyle occurs.

The procedure set up in this work, which was carried out on healthy volunteers without any skin problems, represents the starting point for the evaluation of problematic skin affected by a disease such as acne vulgaris, atopic dermatitis and rosacea, and the efficacy of cosmetic products or the treatment of dysbiosis.

## Figures and Tables

**Figure 1 biomedicines-11-00966-f001:**
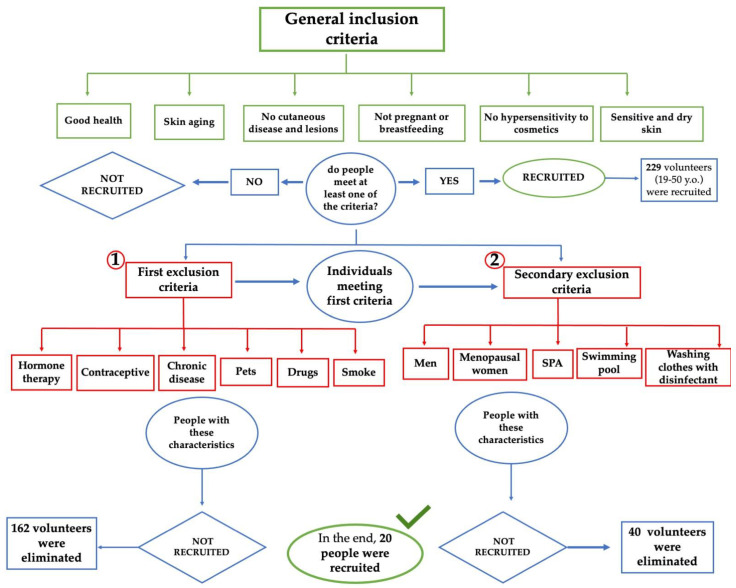
Flow chart of primary (1) and secondary (2) criteria used for the selection of volunteers.

**Figure 2 biomedicines-11-00966-f002:**
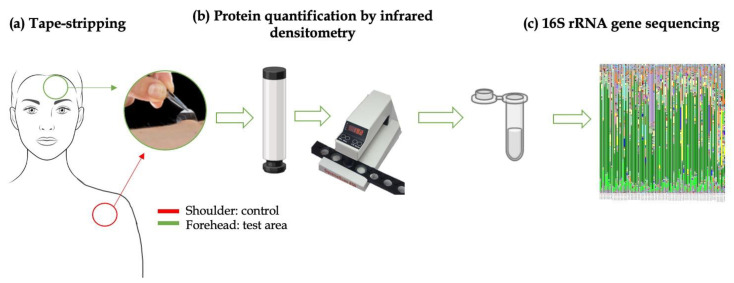
Schematic representation of skin microbiota analysis in in vivo study: (**a**) microbiota collection using tape-stripping on the forehead (green) and shoulder (red) areas; (**b**) protein quantification on tapes collected; (**c**) microbiota analysis on the same tapes collected (figure set up with PowerPoint software).

**Figure 3 biomedicines-11-00966-f003:**
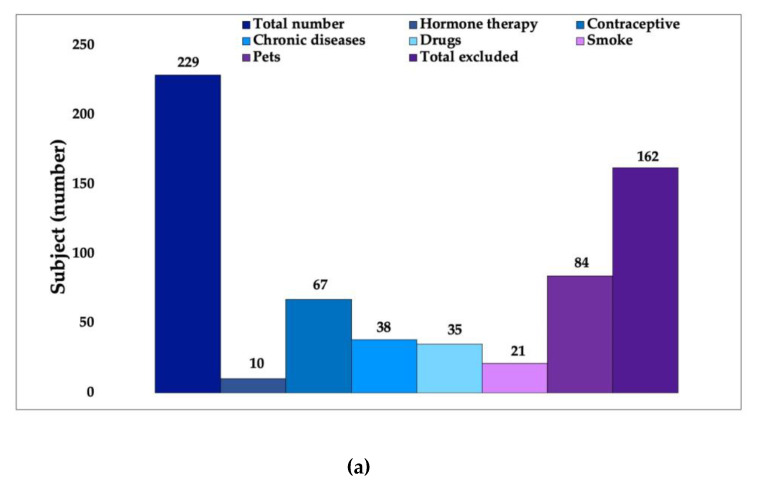
Results from first phase of study concerning the selection of volunteers on the base of a specific questionnaire: (**a**) based on primary criteria from 229 volunteers, 162 people were excluded; (**b**) based on secondary criteria starting from 67 volunteers, 40 subjects were excluded; thus, 27 women were chosen.

**Figure 4 biomedicines-11-00966-f004:**
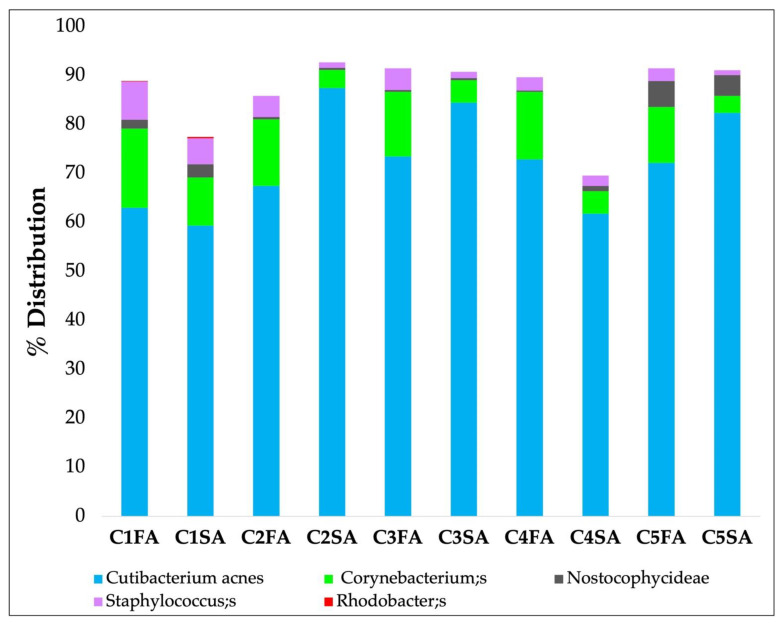
Bacterial species-level profile distribution on 5 strips separately collected and analyzed (from C1 as the outermost layer to C5 as the innermost layer), taken from the forehead A area (FA) and shoulder A area (SA).

**Figure 5 biomedicines-11-00966-f005:**
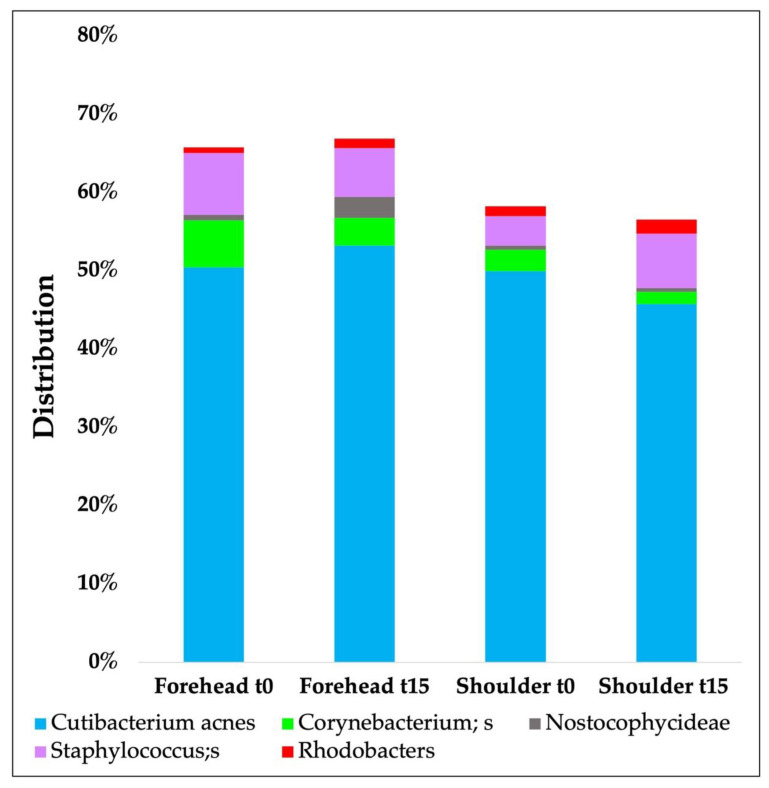
Microbiota analyses results (species level) over time carried out on 20 volunteers.

**Figure 6 biomedicines-11-00966-f006:**
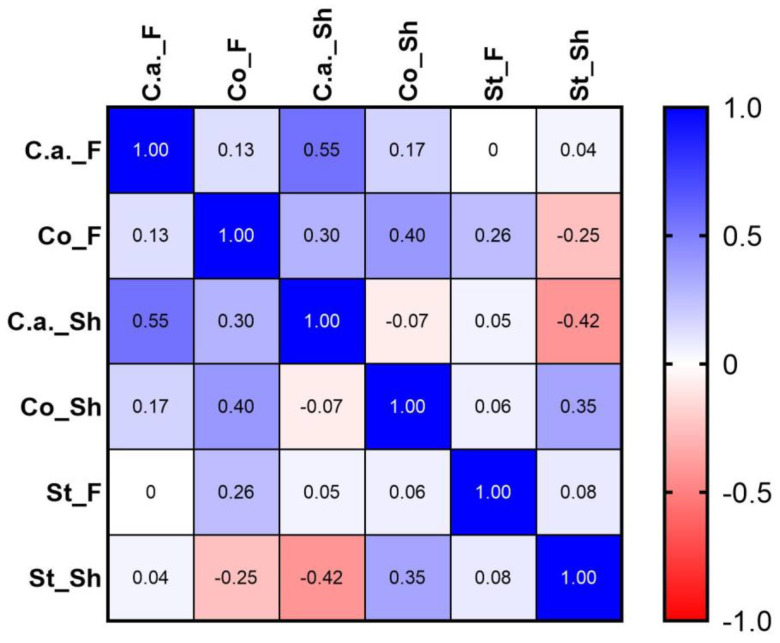
Spearman’s rank correlation coefficients obtained from microbiota analyses results (species level) in the two different areas: F forehead area; Sh shoulder area: C.a = *Cutibacterium acnes*; Co = *Corynebacterium* sp.; St = *Staphylococcus* sp.

**Table 1 biomedicines-11-00966-t001:** Criteria applied for the strength interpretation of Spearman’s rank correlation coefficient.

Coefficient Value		Strength Interpretation
+1	−1	Perfect positive or negative correlation
+0.9–0.7	−0.9–0.7	Very strong correlation
+0.6–0.4	−0.6–0.4	Strong correlation
+0.3	−0.3	Moderate correlation
+0.2	−0.2	Weak correlation
+0.1	−0.1	Negligible correlation
0	0	No correlation

**Table 2 biomedicines-11-00966-t002:** Results concerning instrumental analysis of skin parameters taken at time zero and after 15 days both on the forehead and on shoulder area expressed as a mean and standard deviation. * *p* < 0.05 result is statistically significant; ** *p* < 0.01 is strongly significant.

Parameters	Forehead t0	Forehead t15	Shoulder t0	Shoulder t15	Forehead *p* t0 vs. t15	Shoulder *p* t0 vs. t15
Hydration (A.U.)	53.65	50.22	43.50	43.97	0.0624	0.7984
	6.16	5.98	7.28	5.87		
TEWL (g/m^2^h)	9.74	9.85	8.46	9.31	0.9197	0.0210 *
	1.50	1.09	1.61	1.06		
pH	5.16	5.24	5.53	5.18	0.4358	0.0272 *
	0.43	0.30	0.52	0.32		
Sebum levels (µg/cm^2^)	81.70	99.48	15.85	15.80	0.0054 **	0.3463
	36.35	46.93	9.16	13.59		
Protein content (µg/cm^2)^	15.85	14.76	15.57	14.90	0.2196	0.7490
	3.41	3.64	3.79	3.09		
Porphyrin intensity (A.U.)	186.00	174.15	168.30	170.50	0.0894	0.6017
	19.07	26.28	44.81	24.38		

**Table 3 biomedicines-11-00966-t003:** Spearman’s correlation coefficient comparison and interpretation of the correlation between parameters The value ranges between +1 and −1, where 1 is a total positive correlation and −1 is a total negative correlation.

Parameters	Spearman’s Coefficient	Strength Interpretation
Forehead		
SCWC vs. protein content	−0.69	Strong correlation
Porphyrin intensity vs. sebum	0.44	Strong correlation
Porphyrin intensity vs. protein content	−0.43	Strong correlation
Sebum vs. TEWL	0.43	Strong correlation
Protein content vs. pH	0.32	Moderate correlation
Porphyrin intensity vs. TEWL	0.38	Moderate correlation
Shoulder		
SCWC vs. pH	−0.63	Strong correlation
Porphyrin intensity vs. TEWL	0.40	Strong correlation
Porphyrin intensity vs. sebum	0.48	Strong correlation
Protein content vs. TEWL	0.31	Moderate correlation
Forehead vs. shoulder		
Porphyrin intensity	0.61	Strong correlation
pH	0.46	Strong correlation
TEWL	0.47	Strong correlation

**Table 4 biomedicines-11-00966-t004:** Bacterial phylum level recovered in preliminary analyses with tape strips: B12020 strip; B22020 strip left 24 h in a laboratory environment; B32020 strip after contact with tape stripping device; B42020 as B32020 but also in contact with IR instrument.

Taxonomy	B12020	B22020	B32020	B42020
Bacteria; Firmicutes	9.00	6.20	11.90	15.30
Bacteria; Actinobacteria	70.00	10.90	39.10	26.70
Bacteria; Cyanobacteria	1.60	54.00	0.40	1.00
Bacteria; Proteobacteria	14.40	27.30	41.80	53.40
Bacteria; Bacteroidetes	3.60	1.20	3.90	2.30

**Table 5 biomedicines-11-00966-t005:** Mean bacterial species-level profile distribution obtained by analyzing two strips together (strip 2 and strip 3, C2- 3) both in the forehead area (FA) and the shoulder area (SA).

Taxonomy	C2- 3 FA	C2- 3 SA
Cutibacterium acnes	70.5	86.05
Corynebacterium; s	13.4	4.15
Nostocophycideae	<0.5	<0.5
Staphylococcus; s	4.35	1.2
Rhodobacter; s	<0.5	<0.5

**Table 6 biomedicines-11-00966-t006:** Microbiota analyses results expressed as % distribution at phylum level over time both on forehead and on shoulder areas.

Phyla	Forehead t_0_	Forehead t_15d_	Shoulder t_0_	Shoulder t_15d_
Actinobacteria	62.55 (±20.81)	64.75 (±19.06)	59.30 (±0.21)	59.83 (±27.6)
Proteobacteria	19.62 (±19.52)	16.75 (±12.72)	23.97 (±0.17)	24.53 (±15.56)
Bacteroidetes	2.04 (±2.36)	1.66 (±1.57)	2.44 (±0.02)	2.03 (±1.74)
Cyanobacteria	0.89 (±1.59)	2.86 (±9.33)	1.79 (±0.03)	0.53 (±0.15)
Firmicutes	13.98 (±8.25)	13.03 (±6.42)	11.66 (±0.10)	12.07 (±9.54)

**Table 7 biomedicines-11-00966-t007:** Spearman’s correlation coefficients comparison and interpretation of the correlation between some biophysical parameter and bacterial species. The value ranges between +1 and −1, where 1 is a total positive correlation and −1 is a total negative correlation.

Microbial Species	Spearman’s Coefficient	Strength Interpretation
Forehead		
Porphyrin intensity vs. *Cutibacterium acnes*	0.57	Strong correlation
Porphyrin intensity vs. *Staphylococcus* sp.	−0.33	Moderate correlation
Sebum vs. *Corynebacterium* sp.	0.48	Strong correlation
Sebum vs. *Cutibacterium acnes*	0.40	Strong correlation
pH vs. *Cutibacterium acnes*	0.26	Weak correlation
Protein content vs. *Corynebacterium* sp.	0.45	Strong correlation
Protein content vs. *Staphylococcus* sp.	0.57	Strong correlation
Shoulder		
Sebum vs. *Cutibacterium acnes*	0.41	Strong correlation
pH vs. *Cutibacterium acnes*	0.55	Strong correlation

## Data Availability

The data presented in this study are available in this article.

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
