# Peer review of "Skin Microbiota: Setting up a Protocol to Evaluate a Correlation between the Microbial Flora and Skin Parameters"

_biomedicines, 2023, doi:10.3390/biomedicines11030966_

Round 1

Reviewer 1 Report

Thank you for the opportunity to read the text entitled: Skin Microbiota: Set Up of a Protocol to Evaluate a Correlation between the Microbial Flora and Skin Parameters. The topic taken up is extremely interesting, especially in the context of  half-truths circulating in popular science publications.

Attention is drawn to the very well-written introduction, which shows the scientific background and emphasizes the need for this research. This chapter is well supported by reference literature.

Unfortunately, the following chapters require numerous corrections.

1. keywords: consider whether keywords: bacteria; forehead; shoulder are helpful for the reader, perhaps other keywords would allow for a better description of this work

2. Introduction: "In last period the concept of microbiota is emerging both in pharmaceutical and cosmetic fields and several times in an incorrect way." can you give examples? "Skin (...) the largest organs of human system (5, 6)" - incorrect information. The skin is not the largest organ of the human body. Lines 60: "have joined forces" - such a term is not quite appropriate for scientific paper. Line 91 - enter abbreviation for sebum level since other traits got their abbreviations here. Line 110: "phylotypes (IA1 for example)" - explain what this abbreviation stands for. Line 117 - give sources for the indicated methods, explain the abbreviation NGS.

3. Materials and Methods: Fig 1 is redundant. It does not bring any knowledge and its only advantage is the aesthetic value, which is not necessary in scientific papers. Additionally, all figures and all tables in this manuscript are mislabeled. Tables and figures must comply with the principle of self-description. Taken out of the manuscript, they should be clear and informative to the reader. Therefore, please give precise titles to tables and figures. Please also explain all the abbreviations used in them.

There are many repetitions in the manuscript, e.g. lines 157-159 are redundant, because the indicated information has already been written above. In the next line, it was indicated that the volunteers were women, so why were 5 men then included in this group?

Table 1 is bizarre, mislabeled, and de facto redundant. The following subchapters contain the criteria for inclusion and exclusion from the project.

Line 181: what exactly does skin aging mean?

Line 192 - this is a truism

Lines 202 -232 - the description of the probes used is uneven. Some of them are described in too much detail, some do not even have the name of the product and manufacturer indicated. Please describe all probes in the same way.

Line 257 - check if the indicated sentence does not contain a mistake (typo)

Line 274 - indicate what the purpose was (as for examples A and B)

Section 2.7 needs major revisions. There is no need to indicate literature sources here. It is not indicated what p-value was established as the limit of statistical significance. It was also not indicated in the results section whether the correlations were significant (p for the r coefficient was not shown). How was the type of variables tested? How were the values of skin features on the shoulder and on the forehead compared? On the basis of which source the strength interpretation of correlation coefficient was established. The interpretation indicated here seems to be quite mild, hence such strong results.

4. Results: line 314-316 - repetition. Similarly: 318-326.

Fig. 3 seems uninformative. Consider showing this data in a classic form: a patient flow diagram.

Lines 344-346 - such information is provided in the methodology: the results are shown as means and SD. You don't have to repeat it later.

Line 353 - explain what these strange graphic signs between the numerical values mean.

Table 3 - provide the results of the statistical analysis for the results shown

Lines 371-374 - results are not a chapter to show references. Move it to the discussion. Similarly: 380-381.

Check the manuscript for italics: this is how Latin names should be spelled.

5. Discussion - this manuscript, in fact, contains no discussion. Note that no new reference has been introduced in this chapter. Only one of the introductions was repeated. It is in this chapter that the authors should critically refer to their results and show, on the basis of other publications, the strengths and weaknesses of their project. Please also add a subchapter: study limitation, where the limitations of this work will be enumerated.

6. Conclusions: please re-formulate the conclusions and underline what the most important elements have been learned thanks to this work. What application conclusions can be drawn on its basis?

Lines 498-501 - there is a lack of very important information. Has the project been approved by the bioethics committee? How can I view the results? How was this study and APC funded? Without this information, the work is not ethical.

7. References: numerous errors concerning the bibliometric notation.

Author Response

Dear Reviewer, thank you a lot for your comments. We've changed a lot several part of the paper in order to follow your indications. Please you'll find specific answers to the attached file.

Best regards

Reviewer 2 Report

Authors tried to start up a protocol to correlate as clearly stated in the abstract. However, too much information was provided for some parts. The manuscript is written more as a report style and need to be reformed. Readers would easily get lost due to the disorder. Although statistical analysis was adequately conducted, interpretation is not enough to reveal the science behind.

Introduction

It is informative, but needs to be concise. Some information can be incorporated into methods or discussion. It could be helpful if authors number the objectives in the last paragraph of the introduction.

Methods

2.3 Can be combined with the portion in the introduction (e.g., last paragraph).

2.4 and 2.5 Some part may put in supplementary materials, like “general inclusion criteria”, “general exclusion criteria”, try to make the content concise.

2.6 “Primer sequences used to amplify the variable 16S (V3-V4) and ITS regions are following:”, was these primers used to amplify both the above regions?

2.7 What is the confidence intervals of spearman correlation? Can multiple skin biophysical parameters have the same results on skin microbiota? Interactive effects may be tested.

Results

First paragraph is method.

3.1 The difference between control and treatment should be explained more. What can be seen from Table 3?

3.2 “Cyanobacteria phylum is more present in sample B22020, as it is associated with en-379vironmental bacterial contamination.” How can cyanobacteria be part of the contamination?

Figures

All barplots need to be polished.

Figure 1, More detailed steps number and explanation is needed.

Font size for Figure 3 is small.

Author Response

Dear Reviewer, thank you a lot for your comments. We've changed a lot several part of the paper in order to follow your indications.

Open Review 2

Dear Reviewer,

thank you for your comments.

Herewith below all answers to your requests

Introduction

It is informative, but needs to be concise. Some information can be incorporated into methods or discussion. It could be helpful if authors number the objectives in the last paragraph of the introduction. Done

Methods

2.3 Can be combined with the portion in the introduction (e.g., last paragraph). We’ve modified some parts to be more clear

2.4 and 2.5 Some part may put in supplementary materials, like “general inclusion criteria”, “general exclusion criteria”, try to make the content concise.

2.6 “Primer sequences used to amplify the variable 16S (V3-V4) and ITS regions are following:”, was these primers used to amplify both the above regions? Yes, these primers were used to amplify both regions (V3-V4). In particular primers used are: 16S-341F 5'- CCTACGGGNBGCASCAG -3'

16S-805R 5'- GACTACNVGGGTATCTAATCC -3'. We’ve this specification in the text

2.7 What is the confidence intervals of spearman correlation? Can multiple skin biophysical parameters have the same results on skin microbiota? Interactive effects may be tested.

Results

First paragraph is method. I’m sorry but I disagree because the first paragraph describes the results obtained from the elaboration of 229 questionnaire in order to find a homogeneous panel to perform the following in vivo study

3.1 The difference between control and treatment should be explained more. What can be seen from Table 3? The study was not focused on the evaluation of a treated area vs control. The aim of this part of the work was to find two very similar skin areas in order to set up a protocol for further study (in the future). So forehead will be a test area and shoulder will be the intra-subject control area. Only this kind of protocol can permit to monitor over time the variation of parameters associated to microbiota disorders. So table 3 and Table 6 permitted us to confirm our choice of the two areas. We’ve added explanation into discussion section.

Figures

All barplots need to be polished. Done

Figure 1, More detailed steps number and explanation is needed. We’ve cut this figure

Font size for Figure 3 is small. We’ve changed the Figure

Best regards